# Transcriptome Profile of *Fusarium graminearum* Treated by Putrescine

**DOI:** 10.3390/jof9010060

**Published:** 2022-12-30

**Authors:** Lina Zhang, Xishi Zhou, Pengfeng Li, Yiwei Wang, Qianyong Hu, Yuping Shang, Yunshen Chen, Xiying Zhu, Hongjie Feng, Cuijun Zhang

**Affiliations:** 1Zhengzhou Research Base, State Key Laboratory of Cotton Biology, School of Agricultural Sciences, Zhengzhou University, Zhengzhou 450001, China; 2Shenzhen Branch, Guangdong Laboratory of Lingnan Modern Agriculture, Genome Analysis Laboratory of the Ministry of Agriculture and Rural Affairs, Agricultural Genomics Institute at Shenzhen, Chinese Academy of Agricultural Sciences, Shenzhen 518120, China; 3College of Life Sciences, South China Agricultural University, Guangzhou 510642, China; 4College of Data Science, Taiyuan University of Technology, Taiyuan 030024, China; 5College of Agronomy, Shanxi Agricultural University, Jinzhong 030801, China; 6School of Life Sciences, Henan University, Kaifeng 475004, China; 7Shenzhen Research Institute of Henan University, Shenzhen 518000, China; 8State Key Laboratory of Cotton Biology, Institute of Cotton Research of Chinese Academy of Agricultural Sciences, Anyang 455000, China; 9Western Agricultural Research Center of Chinese Academy of Agricultural Sciences, Changji 831100, China

**Keywords:** deoxynivalenol synthesis, *Fusarium graminearum*, putrescine, transcriptome

## Abstract

*Fusarium graminearum* (*F. graminearum*) is the main pathogen of Fusarium head blight (FHB) in wheat, barley, and corn. Deoxynivalenol (DON), produced by *F. graminearum*, is the most prevalent toxin associated with FHB. The wheat defense compound putrescine can promote DON production during *F. graminearum* infection. However, the underlying mechanisms of putrescine-induced DON synthesis are not well-studied. To investigate the effect of putrescine on the global transcriptional regulation of *F. graminearum*, we treated *F. graminearum* with putrescine and performed RNA deep sequencing. We found that putrescine can largely affect the transcriptome of *F. graminearum*. Gene ontology (GO) and KEGG enrichment analysis revealed that having a large amount of DEGs was associated with ribosome biogenesis, carboxylic acid metabolism, glycolysis/gluconeogenesis, and amino acid metabolism pathways. Co-expression analysis showed that 327 genes had similar expression patterns to *FgTRI* genes and were assigned to the same module. In addition, three transcription factor genes were identified as hub genes in this module, indicating that they may play important roles in DON synthesis. These results provide important clues for further analysis of the molecular mechanisms of putrescine-induced DON synthesis and will facilitate the study of the pathogenic mechanisms of FHB.

## 1. Introduction

Fusarium head blight (FHB), also known as ear rot or scab, is one of the most common wheat diseases in the world [1], causing huge economic losses [2]. Serious FHB occurs every four or five years in particular areas, including China, the USA, and Africa [3]. In recent years, FHB has led to a significant reduction in wheat production [4,5,6,7]. *F. graminearum* and *F. asiaticum* are the main fungi causing FHB [8]. In addition to wheat, *F. graminearum* can also cause FHB on other grains, such as barley and corn [9].

*F. graminearum* can produce large amounts of trichothecene mycotoxins, which are seriously toxic to humans and animals [10], including deoxynivalenol (DON), nivalenol (NIV), and zearalenone (ZEA) [11,12]. DON can cause vomiting, diarrhea, oxidative damage, and digestive tract disorders in humans and animals [13,14]. It can also cause biological barriers and affect the function and viability of cells and organs [15]. DON can bind to ribosomal peptide transferase and activate cytokines to inhibit nucleic acid and protein biosynthesis [16]. Almost all food crops, including wheat, corn, barley, and others that are infected by *F. graminearum* contain DON [17]. DON is also a vital virulence factor of *F. graminearum* [18]. Exploring the regulation mechanism of DON synthesis is of great significance to the study of the pathogenic mechanisms and control of FHB. Previous studies have shown that putrescine synthesized by plants after *F. graminearum* infection can promote the production of DON [19,20]. However, the underlying mechanisms of putrescine-induced DON synthesis in *F. graminearum*, especially at the transcriptome level, are not well-understood.

With the development of high-throughput sequencing technology, RNA deep sequencing plays an important role in the study of transcriptome profiles of organisms and is an effective tool for mechanism research. In this study, we conducted a transcriptome analysis of *F. graminearum* under exogenous putrescine treatment at different time points to systematically investigate the molecular mechanism of putrescine-induced DON production. We identified 5256, 5498, 3850, and 3002 differentially expressed genes (DEGs) in samples treated with putrescine for 3 h, 6 h, 12 h, and 24 h, respectively. Gene ontology (GO) and KEGG enrichment analysis revealed that many DEGs were associated with ribosome biogenesis, carboxylic acid metabolism, glycolysis/gluconeogenesis, and amino acid metabolism pathways. We obtained six gene clusters through expression pattern analysis. Interestingly, we found that all 14 *FgTRI* genes were classified into cluster five and the genes in cluster five were monotonously downregulated after 3 h of putrescine treatment and then continuously upregulated at 12 h and 24 h. Co-expression analysis revealed that 327 genes had similar expression patterns to *FgTRI* genes and were assigned to the same module. In addition, three transcription factor genes were identified as hub genes in this module, indicating that they may play important roles in DON synthesis. This study is the first comprehensive transcription profiling of *F. graminearum* treated by putrescine, and its results provide valuable support for further studies to elucidate the molecular mechanisms of DON synthesis.

## 2. Materials and Methods

### 2.1. F. graminearum Strain and Growth Conditions

The strain used in this experiment was *F. graminearum* strain PH-1, which was kindly provided by Dr Huiquan Liu at Northwest A & F University (Yangling, China). *F. graminearum* was routinely cultured in potato dextrose agar (PDA) medium at 25 °C for 7 days. Five 100 mL bottles of liquid potato dextrose broth (PDB) medium with 5 mm freshly grown mycelia taken from the edge of a colony were shaken for 2–3 days. The hyphae were collected and washed once with double-distilled water individually and then transferred into 5 corresponding bottles of liquid trichothecene biosynthesis induction (TBI) medium (87.64 mM sucrose, 5 mM arginine, 7.35 mM KH_2_PO_4_, 2.08 mM MgSO_4_, 6.71 mM KCL, 40.57 mM MgSO_4_·7H_2_O, 0.3 g/L plant gel, 0.2 mL trace element, and distilled H_2_O up to 1000 mL) for 24 h. One bottle of hyphae was frozen in liquid nitrogen as the 0 h sample. Sterilized putrescine (Sigma Aldrich (Shanghai) Trading Co., Ltd., Shanghai, China) was added to the remaining 4 bottles of TBI medium to a 5 mM final concentration [21], and hyphae were shaken at 28 °C for 3 h, 6 h, 12 h, and 24 h. Finally, the hyphae of *F. graminearum* cultured at different time points were collected and frozen in liquid nitrogen for total RNA extraction. Three biological replicates were set at each time point.

### 2.2. RNA Extraction, Library Preparation, and Sequencing

For RNA-seq analysis, the frozen samples were ground into fine powder in a mortar with a pestle in the presence of liquid nitrogen. Total RNA was extracted using an RNAprep Pure Plant Plus Kit according to the manufacturer’s instructions, and the concentration of RNA was estimated using a NanoDrop 2000c ultra-trace biological detector. The total RNA samples were sent to BGI for library construction and sequencing. Libraries were constructed using an MGIEasy RNA Library Preparation Kit and sequenced on the DNBseq platform using a paired-end scheme.

### 2.3. Quality Control and Data Assembly

SOAPnuke was used to obtain high-quality clean data by removing adapter sequences and low-quality reads [22]. The Q20 and GC content were evaluated after quality control of clean reads using FastQC software. All clean data were mapped to the reference genome of *F. graminearum* using hisat2, and unique mapping reads were identified.

### 2.4. Analysis of DEGs and Functional Annotation

The gene expression level was assessed as fragments per kilobase of transcripts per million reads mapped (FPKM). The reads were counted by featureCounts software, and the differential expression analysis of 5 inoculation time points was conducted by the DESeq2 R software package. Genes with *p*-value < 0.05 and |log_2_ (fold change)| > 1 were designated as differentially expressed genes (DEGs). Gene ontology (GO) and Kyoto Encyclopedia of Genes and Genomes (KEGG) enrichment analysis of DEGs was carried out using the clusterProfiler R software package [23]. The GO function and KEGG pathway were considered enriched with a *p*-value of 0.05 as the significance threshold. Potential transcription factors (TFs) were identified by the Fungal Transcription Factor Database (FTFD) with default parameters (http://ftfd.snu.ac.kr/index.php?a=view, accessed on 26 February 2008).

### 2.5. Weighted Gene Co-Expression Network Analysis (WGCNA)

To detect co-expression modules and key regulatory genes associated with DON production in *F. graminearum* under putrescine treatment, we generated co-expression networks using the WGCNA package in R as previously described [24]. Briefly, only expressed genes with average FPKM values higher than 1 in any sample were further processed. The soft thresholding power was determined using the pickSoftThreshold function based on the scale-free topology model fit (R^2^) > 0.8. Then, the automatic blockwiseModules network construction approach was applied to obtain highly correlated modules. The *FgTRI* gene module was defined as the key module, and the co-expression and transcriptional regulatory networks were displayed using Cytoscape v3.9.1 [25].

## 3. Results

### 3.1. Evaluation of Transcriptome Sequencing Data

In order to identify the key genes regulating the production of DON under putrescine treatment, hyphae samples were collected for RNA deep sequencing at five treatment time points (0 h, 3 h, 6 h, 12 h, and 24 h). The GC content of all samples ranged from 51.18 to 53.88%, and Q20 scores ranged from 95.86 to 97.07%. Approximately 96.61 to 97.32% of clean reads were successfully mapped to the reference genome. Furthermore, 95.34 to 96.46% of clean reads were uniquely mapped (Table 1). The evaluation results indicated that the quality of transcriptome sequencing was sufficient for further expression analysis.

According to principal component analysis (PCA) and correlation analysis, we removed sample 12 h-3 because it showed a high deviation from the other two replicates (Figure 1). The biological replicates of samples at other time points were highly consistent with each other (R^2^ > 0.95) (Figure 1B). The correlation coefficient between 6 h and 12 h sample groups was very low (R^2^ < 0.4), suggesting obvious transcriptome changes between these two time points (Figure 1B).

### 3.2. Differential Gene Expression Analysis

Compared with 0 h samples, 5256, 5498, 3850, and 3002 DEGs were identified in samples treated with putrescine for 3 h, 6 h, 12 h, and 24 h, respectively, and more genes were downregulated than upregulated at each time point (Figure 2). The number of DEGs increased in the 6 h treatment (5498) and then decreased at 12 h (3850), and decreased further at 24 h (3002) (Figure 2). Differential expression analysis of comparison groups of adjacent time points was also performed. We identified the most DEGs (3891) in the comparison of 6 h vs. 12 h and the least (1202) in 3 h vs. 6 h (Figure 2). These results indicated that 6 h—12 h is probably the most important stage for transcriptome changes of *F. graminearum* during putrescine treatment.

### 3.3. GO Enrichment Analysis of DEGs

In order to screen the unique insights on the effect of putrescine treatment on the biological process of *F. graminearum*, GO enrichment analysis was carried out on DEGs at four time points: 3, 6, 12, and 24 h. In order to highlight the ontologies and pathways relevant to this study, subsets of highly enriched GO terms were selected (Figure 3). DEGs identified in the 3 h and 6 h putrescine treatment groups were enriched in many of the same pathways, including translation, peptide biosynthetic process, and ribosome-related process (Figure 3). After 12 h of putrescine treatment, DEGs were mostly enriched in the nucleolus, ribosome biogenesis, and carbohydrate derivative biosynthetic process (Figure 3). After 24 h, the DEGs enrichment terms were further changed to small molecule metabolic process, organic acid metabolic process, and aromatic amino acid family catabolic process (Figure 3). Notably, ribosome and ribonucleoprotein complex biogenesis pathways were significantly enriched in DEGs at 3 h, 6 h, and 12 h (Figure 3). In addition, the amino acid catabolic process pathway was significantly enriched in DEGs at 24 h (Figure 3), indicating that ribosome, ribonucleoprotein, and amino acid catabolic pathways play important roles in the response of *F. graminearum* in the early stage (<24 h) of putrescine treatment.

### 3.4. KEGG Enrichment Analysis of DEGs

In order to further determine the specific metabolic pathway of *F. graminearum* under putrescine treatment, a KEGG analysis of DEGs was performed. A total of 825, 718, 589, and 318 DEGs were enriched in different metabolic pathways at 3 h, 6 h, 12 h, and 24 h, respectively. In order to highlight the pathways relevant to this study, we selected the pathways of interest that showed significant enrichment (Figure 4). DEGs identified in the 3 h and 6 h putrescine treatment groups were enriched in many of the same pathways, including nucleocytoplasmic transport and amino acid biosynthesis pathways (Figure 4). After 12 h of putrescine treatment, DEGs were mostly enriched in glycolysis/gluconeogenesis, galactose metabolism, and amino acid metabolism pathways (Figure 4). After 24 h, the main DEG enrichment terms were nitrogen metabolism, tryptophan metabolism, and amino acid metabolism (Figure 4). Combined with the results of GO analysis (Figure 3), DEGs identified at all treatment time points were enriched in the common ribosome and amino acid metabolism pathways, again indicating that ribosome and amino acid metabolism play important roles in the early stage of putrescine treatment in *F. graminearum*. In addition, we found that the glycolysis/gluconeogenesis pathway was significantly enriched in the upregulated DEGs only after 12 h and 24 h of putrescine treatment (Figure 4), and the CoA biosynthesis pathway was enriched in upregulated DEGs after 12 h of treatment (Figure 4). Acetyl-CoA, which is produced by glycolysis, is an intermediate substance of the DON biosynthesis process [26]. Thus, we speculated that putrescine regulates DON synthesis by influencing ribosomal functions, amino acid metabolism, and the glycolysis/gluconeogenesis process.

### 3.5. Identification of Differentially Expressed TFs in F. graminearum Treated by Putrescine

Transcription factors are key molecules in the regulation of gene expression and play an important role in dealing with various stresses. In this study, 69 consistently differentially expressed TFs belonging to eight transcription factor families were detected from *F. graminearum* treated for 3 h, 6 h, 12 h, and 24 h (Figure 5A). A total of 14 differently expressed TFs belonging to six TF families were identified at all treatment time points compared to 0 h. Among them, the most abundant TF family was Zn2Cys6 (6), followed by C2H2 zinc finger (3), HMG (2), bHLH (1), nucleic acid-binding, OB-fold (1), and bZIP (1) (Figure 5B).

### 3.6. Expression Pattern Analysis of All Genes under Putrescine Treatment

We used the mfuzz package in R [27] (http://itb1.biologie.hu-berlin.de/~futschik/software/R/Mfuzz/index.html, accessed on 20 May 2007) to analyze the different gene expression patterns based on all of the genes (13,285; FPKM < 1 filtered out). We obtained six gene clusters with different expression patterns, with 1952, 2438, 1904, 2689, 1790, and 2511 genes in clusters 1–6, respectively (Figure 6A,B; Appendix A). Interestingly, we found that all 14 *FgTRI* genes were classified into cluster five and the genes in cluster five were monotonously downregulated after 3 h of putrescine treatment and then continuously upregulated at 12 h and 24 h (Figure 6A), indicating that the production of DON under putrescine treatment started after 6 h treatment.

### 3.7. Construction of Co-Expression Networks

To identify the co-expression networks associated with DON biosynthesis, we used the WGCNA R software package based on the gene FPKM matrix (13,285; FPKM < 1 filtered out). The sample clustering analysis revealed strong repeatability among the biological replicates (Appendix A). In addition, we found that the 3 h and 6 h samples were grouped together, while the 0 h, 12 h, and 24 h samples formed a second group, which again indicated that there were significant transcriptional changes between 6 h and 12 h during putrescine treatment of *F. graminearum* (Appendix A). In the WGCNA pipeline, based on the scale-free topology criterion with R^2^ = 0.9, we set the soft threshold as 18 (Appendix A). Then, we used the automatic blockwiseModules network construction approach to identify co-expression modules. A color scheme was used to allow visualization of modules by showing highly correlated genes in the same color (Appendix A). We obtained a total of 23 color modules, each composed of genes with similar expression patterns over time (Appendix A). The relatively low correlation coefficients between pairwise color modules showed that our functional color modules were clearly divided (Appendix A). Notably, *FgTRI* genes, except *FGSG_03538*, were assigned to the same MEblack module (Appendix A), which is consistent with the result of the mfuzz analysis. We then displayed the co-expression network of the MEblack module using Cytoscape software. Three TF genes, *FGSG_01915* (Myb), *FGSG_03292* (Zn2Cys6), and *FGSG_03536* (C2H2 zinc finger), were identified as hub genes in the MEblack module. They were co-expressed with 12, 12, and 7 *FgTRI* genes, respectively, indicating that these *FgTRI* genes may be the downstream targets of the three hub TFs (Figure 7).

## 4. Discussion

### 4.1. Expression of FgTRI Genes Began to Increase after 6 h of Putrescine Treatment

Previous studies have shown that putrescine can induce toxin DON production in *F. graminearum* [28]; however, the detailed molecular mechanism remains unclear. We compared the number of DEGs of *F. graminearum* at different treatment time points relative to 0 h samples and found that the number increased at 6 h compared to 3 h, then continually decreased at 12 h and 24 h (Figure 2). The most DEGs were found in the comparison of 6 h vs. 12 h (Figure 2), and there was a lower correlation coefficient between these two time points according to the heat map results (Figure 1B), indicating that 6 h to 12 h was the most significant period for *F. graminearum* transcriptional changes under putrescine treatment.

The deoxynivalenol (DON) biosynthesis process involves oxidation, esterification, and isomerization, using farnesyl pyrophosphate (FPP) as the substrate [29,30], and *FgTRI* genes play a crucial role in this process [31]. It has been reported that when *F. graminearum* infects wheat, the polyamine synthesis pathway is activated, in which the representative polyamine putrescine, a plant stress response substance, accumulates in plants. The accumulated putrescine activates the nucleation of transcription factor *FgAreA* in *F. graminearum,* resulting in the activation of *FgTRI* gene transcription and the promotion of DON synthesis [28]. Previous studies found that *FgTRI* gene expression was elevated in *F. graminearum* treated with putrescine for 48 h [28]. However, comparing RNA-seq data at different time points in the early stage (<24 h) of putrescine treatment, we found that after 3 h of treatment, the expression of all *FgTRI* genes decreased significantly, and expression began to increase after 6 h and 12 h, then increased significantly after 24 h of treatment, indicating that putrescine could rapidly increase *FgTRI* gene expression in the early stage of treatment (Figure 6A and Figure 8B).

### 4.2. Effect of Amino Acid Biosynthesis and Metabolism on DON Production

According to the results of GO and KEGG enrichment analysis, the DEGs identified in *F. graminearum* treated with putrescine were mainly involved in the glycolysis/gluconeogenesis pathway, amino acid metabolism, and carbohydrate metabolism. Notably, DEGs identified at all time points were commonly enriched in amino acid metabolism pathways (Figure 4). Previous studies have shown that trichothecene production is influenced by certain amino acids through the activation of *FgAreA* expression and increased *FgTRI* gene transcription [32]. *FgAreA* can use L-Asp, L-Cys, Gly, L-Glu, L-His, L-Ile, L-Leu, L-Lys, L-Thr, L-Trp, L-Tyr, and L-Val to regulate DON synthesis in *F. graminearum* [33]. We found that DEGs participating in beta-alanine metabolism, arginine and proline metabolism, tryptophan metabolism, valine, leucine, and isoleucine degradation, and tyrosine metabolism pathways were downregulated in the earlier 3 h and 6 h treatments (Figure 4) and upregulated in the later 12 h and 24 h treatments (Figure 4). The significant changes in the expression of genes involved in amino acid biosynthesis and metabolism indicate that amino acid metabolism plays an important role in putrescine-induced DON toxin synthesis.

### 4.3. Effect of Glycolysis on DON Production

DON toxin is synthesized using FPP as the substrate through a series of reactions (Figure 8A). FPP is synthesized from acetyl-CoA, which is produced by glycolysis using glucose as the starting substance [34]. Therefore, glycolysis plays an important role in DON synthesis. According to our KEGG results, the DEGs upregulated at 12 h and 24 h were highly enriched in the glycolysis/gluconeogenesis pathway, and those upregulated at 12 h were highly enriched in the CoA biosynthesis pathway (Figure 4). *FgNTH* (*FGSG_09895*) encoding trehalase was found [35], which mediates the conversion of trehalose to glucose in filamentous fungi [18]. In our results, *FgNTH* was found to be upregulated to varying degrees at all four time points of putrescine treatment compared to 0 h (Figure 8B). We, therefore, hypothesize that putrescine can induce DON synthesis by promoting glucose production through the stimulation of trehalase expression in *F. graminearum*.

### 4.4. FGSG_01915, FGSG_03292, and FGSG_03536 May Be Key TF Genes Involved in DON Synthesis in F. graminearum

Our WGCNA identified the MEblack module that contained the most *FgTRI* genes (except *FgTRI10*, *FGSG_03538*) and other genes with the same expression pattern. Three TF genes, *FGSG_01915* (Myb), *FGSG_03292* (Zn2Cys6), and *FGSG_03536* (C2H2 zinc finger), were identified as hub genes in the MEblack module. *FGSG_03536* (*FgTRI6*), an *FgTRI* gene that encodes a protein belonging to the C2H2 zinc finger transcription factor family, is a major transcriptional regulator of trichothecene production and plays an important role in DON synthesis [36,37]. Consistently, our results show that *FGSG_03536* was co-expressed with seven *FgTRI* genes (Figure 7C). In addition, *FGSG_01915* was co-expressed with 12 *FgTRI* genes (Figure 7A). However, the role of *FGSG_01915* in DON synthesis is still unknown and needs further study. *FGSG_03292*, which belongs to the Zn2Cys6 transcription factor family, has a regulatory role in the production of 3-acetyldeoxynivalenol (3ADON) in *F. graminearum* [38], and our results show that it was co-expressed with 12 *FgTRI* genes (Figure 7B). Therefore, we speculate that *FGSG_03292* may also have a role in DON synthesis. According to our RNA-seq results, the expression levels of these three transcription factors began to increase after 6 h of putrescine treatment and reached the highest level at 24 h (Appendix A), consistent with the expression pattern of *FgTRI* genes (Figure 8A). In summary, we suggest that putrescine treatment activates the expression of transcription factors *FGSG_01915*, *FGSG_03292*, and *FGSG_03536*, promoting the transcription of *FgTRI* genes and ultimately leading to the production of DON.

## 5. Conclusions

In this study, by means of RNA-seq, we found that putrescine could largely affect the transcriptome of *F. graminearum*. The significant changes in the expression of genes involved in carboxylic acid metabolism, glycolysis/gluconeogenesis, and amino acid metabolism pathways indicate that these pathways play important roles in putrescine-induced DON toxin synthesis. Co-expression analysis revealed that 327 genes had similar expression patterns to *FgTRI* genes and were assigned to the same module. Further functional study of the co-expressed 327 genes will provide important insights into the molecular mechanisms of DON synthesis.

## Figures and Tables

**Figure 1 jof-09-00060-f001:**
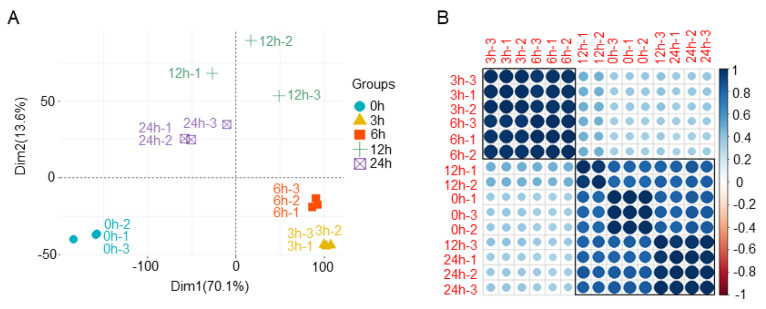
Sample reproducibility and reproducibility of *F. graminearum* after putrescine treatment. (**A**) Principal component analysis (PCA) shows distances between samples of transcripts from different *F. graminearum* treatment groups. (**B**) Heat map of RNA expression values between treatment groups. Pearson correlation coefficients are represented by color and size.

**Figure 2 jof-09-00060-f002:**
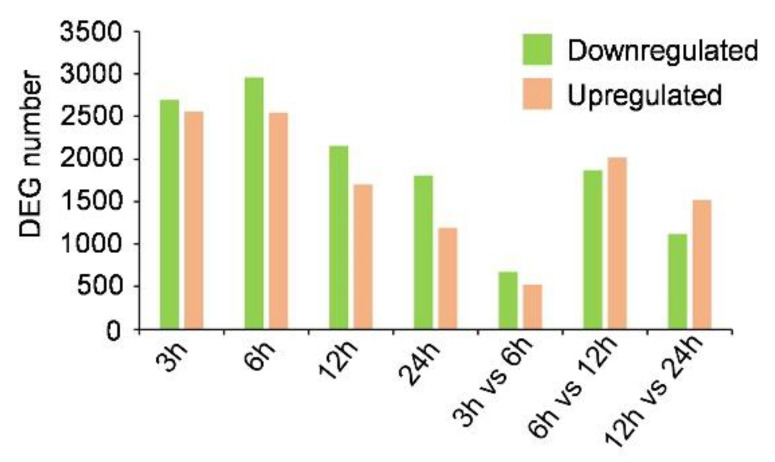
Bar graph showing number of DEGs of *F. graminearum* at different time points or adjacent time periods after putrescine treatment.

**Figure 3 jof-09-00060-f003:**
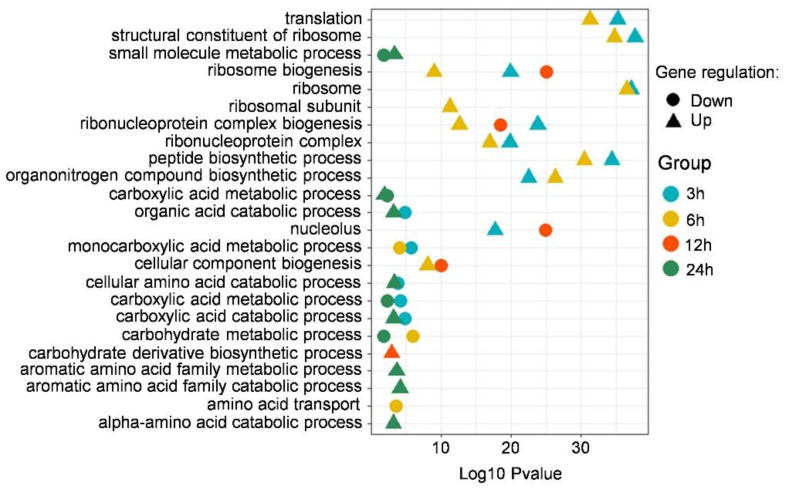
Specified GO analysis of DEGs upon putrescine treatment. Subsets of categorized GO terms identified with DEGs from putrescine treatment are presented in each panel. Colors represent time points: blue, 3 h; orange, 6 h; red, 12 h; green, 24 h.

**Figure 4 jof-09-00060-f004:**
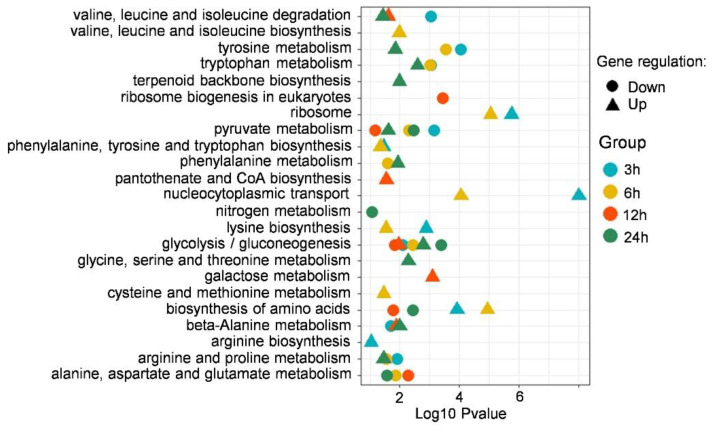
Specified KEGG analysis of DEGs identified in comparison groups. Colors represent treatment time points: blue, 3 h; orange, 6 h; red, 12 h; green, 24 h.

**Figure 5 jof-09-00060-f005:**
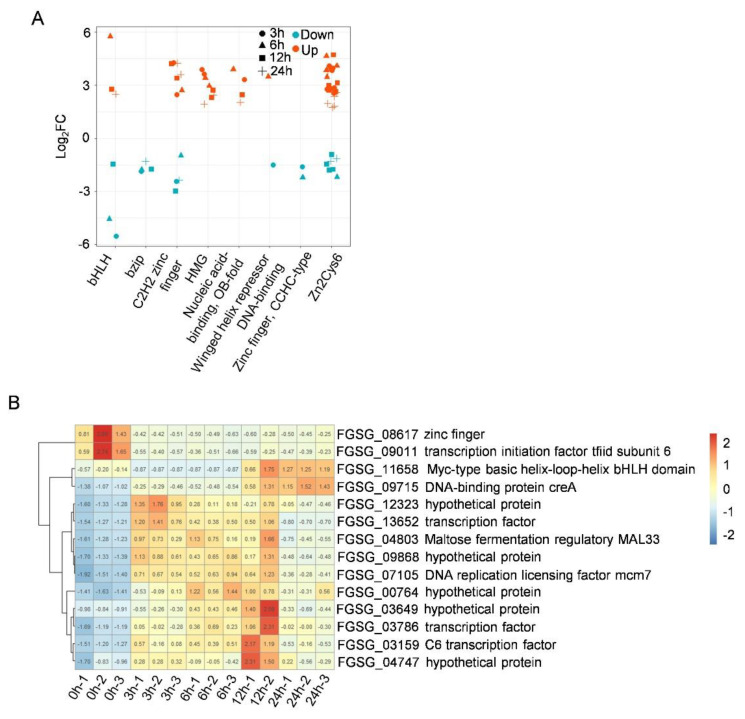
Identification and distribution of transcription factors. (**A**) Scatter plot showing transcript levels (Log2 FC) of consistently differentially expressed transcription factors at all time points (3 h, 6 h, 12 h, and 24 h) compared to 0 h. X-axis represents TF family, in which identified TFs are located. (**B**) Heat map illustrating transcript levels of 14 differentially expressed transcription factors after putrescine treatment for 3 h, 6 h, 12 h and 24 h. Color scale indicates normalized FPKM values (log2 FPKM).

**Figure 6 jof-09-00060-f006:**
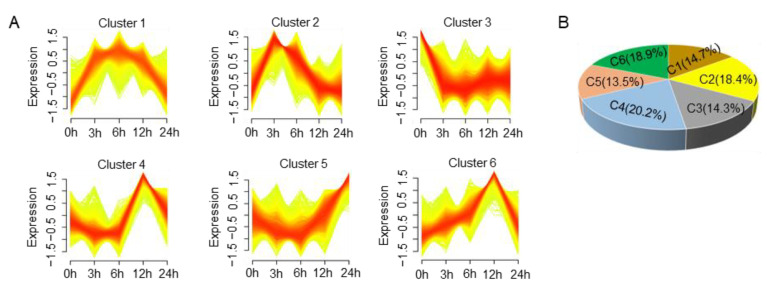
Dynamic changes of *F. graminearum* gene expression during putrescine treatment. (**A**) Six gene expression patterns identified by mfuzz c-means clustering analysis. X-axis represents treatment time points, and Y-axis represents log2 transformation and normalization intensity ratio of each stage. (**B**) Pie chart showing percentage of genes in each cluster.

**Figure 7 jof-09-00060-f007:**
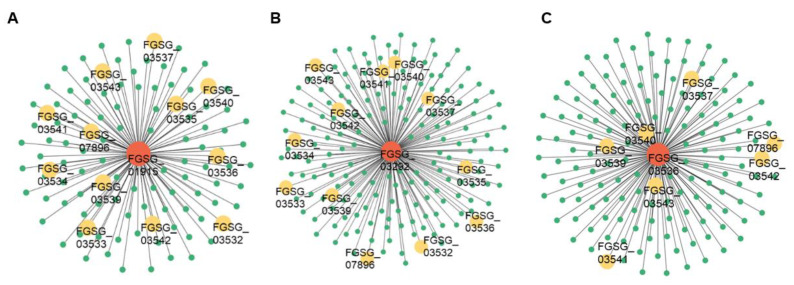
Co-expression networks of (**A**) *FGSG_01915*, (**B**) *FGSG_03292*, and (**C**) *FGSG_03536* and neighboring genes during putrescine treatment. Red represents hub genes, yellow represents *FgTRI* genes, green represents other genes in MEblack module.

**Figure 8 jof-09-00060-f008:**
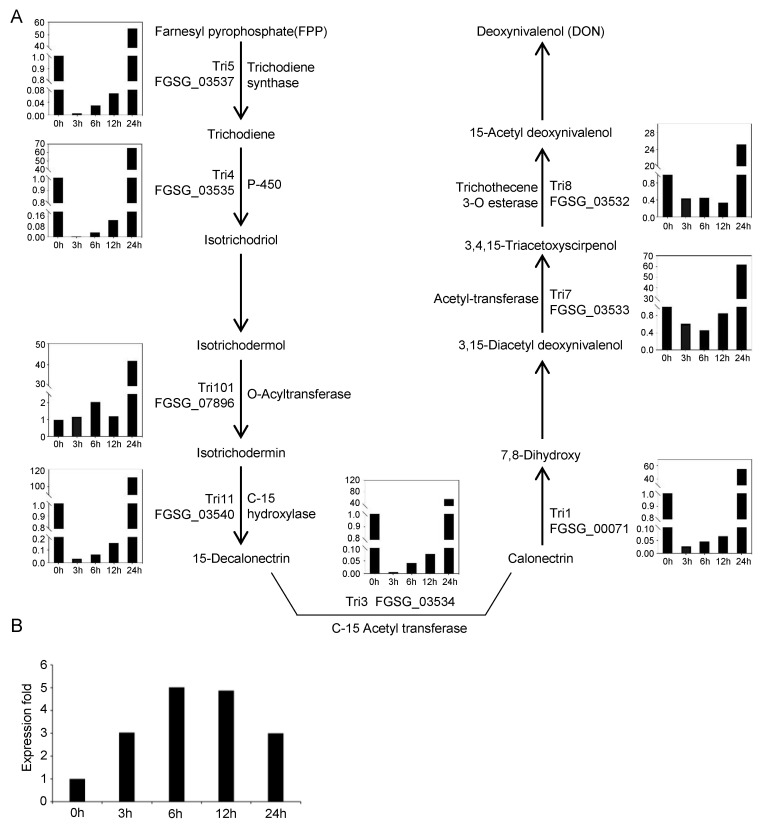
Biosynthetic pathway of DON in *F. graminearum* and expression of key genes of DON synthesis. (**A**) DON biosynthetic pathway and related gene expression. Gene expression levels are from RNA-seq data. (**B**) Dynamic fold expression of *FgNTH* (*FGSG_09895*) with processing time.

**Table 1 jof-09-00060-t001:** Summary of statistics for sequence quality control and genome mapping.

Samples	Clean Data			Mapped Reads		
	Clean Reads	Q20 (%)	GC (%)	Total Reads	Mapped Reads (%)	Unique Mapped Reads (%)
0 h	22,479,084	96.12	52.54	21,130,739	20,424,795 (96.66%)	20,301,372 (96.08%)
0 h	24,042,002	96.45	52.50	48,283,839	46,810,669 (96.95%)	46,481,429 (96.27%)
0 h	24,924,258	97.07	52.43	50,074,181	48,618,073 (97.09%)	48,234,367 (96.33%)
3 h	24,404,822	96.98	53.83	49,178,115	47,662,961 (96.92%)	47,058,097 (95.69%)
3 h	24,083,702	96.12	53.82	48,540,560	46,935,242 (96.69%)	46,326,950 (95.44%)
3 h	22,092,868	96.14	53.73	44,533,167	43,027,195 (96.62%)	42,459,736 (95.34%)
6 h	23,796,099	96.94	53.88	47,915,912	46,596,506 (97.25%)	46,045,762 (96.10%)
6 h	23,784,079	95.86	53.75	47,893,717	46,268,262 (96.61%)	45,718,258 (95.46%)
6 h	24,207,815	97.06	53.12	48,760,129	47,454,344 (97.32%)	46,919,058 (96.22%)
12 h	22,677,537	96.69	51.18	45,602,204	44,365,857 (97.29%)	43,988,722 (96.46%)
12 h	24,108,403	96.28	52.57	48,460,537	46,979,214 (96.94%)	46,566,522 (96.10%)
12 h	22,341,753	96.19	51.91	44,895,324	43,460,348 (96.80%)	43,102,193 (96.01%)
24 h	22,892,156	96.48	51.26	45,998,189	44,688,697 (97.15%)	44,351,292 (96.42%)
24 h	22,775,413	96.33	51.52	45,736,758	44,387,923 (97.05%)	44,078,173 (96.37%)
24 h	22,113,432	96.09	51.70	44,418,523	43,042,789 (96.90%)	42,712,534 (96.16%)

## Data Availability

The data presented in this study were deposited in the National Center for Biotechnology Information (NCBI) database (PRJNA891915).

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
