# Peer review of "Transcriptome Profile of *Fusarium graminearum* Treated by Putrescine"

_jof, 2022, doi:10.3390/jof9010060_

Round 1
Reviewer 1 Report
I carefully read the manuscript "Transcriptome Profile of Fusarium graminearum Treated by Putrescine". I believe that the manuscript's contents are very interesting and fit the scope of the journal. Overall, from hypothesis to conclusion, the study is compact and well presented.
Minor Comments
- Add clearly research background problem and their possible solution in the introduction section.
- Mention the statistical analysis methods.
- Line-311- Mention the future study properly.
Major Comment
I could not find the Figures throughout the manuscript. The authors should provide all the figures before publication.
Author Response
Reviewer 1
Comments and Suggestions for Authors
I carefully read the manuscript "Transcriptome Profile of Fusarium graminearum Treated by Putrescine". I believe that the manuscript's contents are very interesting and fit the scope of the journal. Overall, from hypothesis to conclusion, the study is compact and well presented.
We thank this reviewer’s positive comments.
Minor Comments
- Add clearly research background problem and their possible solution in the introduction section.
As suggested, the research background problem and the possible solution have been included in the introduction section. And the last two paragraphs of the Introduction have been reorganized.
The sentence “However, the underlying mechanisms of putrescine-induced DON synthesis in F. graminearum, especially at the transcriptome level, are not well-understood” describes the research background problem.
The sentence “In this study, we conducted a transcriptome analysis of F. graminearum under exogenous putrescine treatment at different time points to systematically investigate the molecular mechanism of putrescine-induced DON production” describes the possible solution.
- Mention the statistical analysis methods.
As suggested, we have added the statistical analysis methods in the Materials and Methods section. For example, we have included the sentence “The GO function and KEGG pathway were considered enriched with the P-value of 0.05 as the significance threshold” in methods section 2.4. In addition, we have described the Quality control and data assembly analysis in more detail in the methods section 2.3.
- Line-311- Mention the future study properly.
We thank the reviewer for this suggestion. We have included the sentence “Co-expression analysis revealed that 327 genes had similar expression patterns to FgTRI genes and were assigned to the same module. Further functional study of the co-expressed 327 genes will provide important insights into the molecular mechanisms of DON synthesis.” in our revised manuscript to introduce the future study.
Major Comment
- I could not find the Figures throughout the manuscript. The authors should provide all the figures before publication.
We thank the reviewer for notice us this. We have added all the figures in the revised manuscript.

Reviewer 2 Report
In this study, the authors found by RNA-seq analysis that putrescine can largely affect the transcriptome of F. graminearum. Genes were significantly altered in expression. The significant changes in the expression of genes involved in carboxylic acid metabolism, glycolysis/gluconeogenesis, and amino acid metabolism pathways indicate that these pathways play important roles in putrescine-induced DON toxin synthesis. Moreover, the figure mentioned in this manuscript was not attached. So unable to review it completely.
The experimental results are interesting, but some comments are as follows:
1. In lines 29,72, 66, 91, and 131"3 h, 6 h ......" note the formatting and the need for spaces between numbers and units. There are many such errors in the article and the author is requested to check the manuscript carefully.
2. Please write the source of the strain (F. graminearum) in the Materials and Methods section.
3. There is an error in the tense of the sentence in line 149 and the writer needs to double-check the tense and grammar of the text as well as the formatting of the writing. Try to keep the sentenses consistent throughout the text.
4. Nearly half of the references in this article are more than or nearly a decade old. It is recommended that the author incorporate more new research in their citations and discussions.
5. It is recommended that the authors add a description of the novelty of the article in the introduction and abstract sections.
The experimental notes in this article are overly interesting, but the novelty of the article is not adequately described and the manuscript writing suffers from grammatical and formatting errors.
Author Response
Reviewer 2
Comments and Suggestions for Authors
In this study, the authors found by RNA-seq analysis that putrescine can largely affect the transcriptome of F. graminearum. Genes were significantly altered in expression. The significant changes in the expression of genes involved in carboxylic acid metabolism, glycolysis/gluconeogenesis, and amino acid metabolism pathways indicate that these pathways play important roles in putrescine-induced DON toxin synthesis.
We thank the reviewer for the positive comments to our manuscript.
Moreover, the figure mentioned in this manuscript was not attached. So unable to review it completely.
We thank the reviewer for notice us this. We have added all the figures in the revised manuscript.
The experimental results are interesting, but some comments are as follows:
- In lines 29,72, 66, 91, and 131"3 h, 6 h ......" note the formatting and the need for spaces between numbers and units. There are many such errors in the article and the author is requested to check the manuscript carefully.
We thank the reviewer for notice us this. We have fixed all of them.
- Please write the source of the strain (F. graminearum) in the Materials and Methods section.
The strain used in this experiment was F. graminearum strain PH-1, which was kindly provided by Dr Huiquan Liu at Northwest A&F University (Yangling, China). We have included this in the revised manuscript.
- There is an error in the tense of the sentence in line 149 and the writer needs to double-check the tense and grammar of the text as well as the formatting of the writing. Try to keep the sentenses consistent throughout the text.
We thank the reviewer for notice us this. We have fixed it.
- Nearly half of the references in this article are more than or nearly a decade old. It is recommended that the author incorporate more new research in their citations and discussions.
As suggested, we have gone through the references carefully and incorporated 8 new research paper. In addition, we have deleted 2 paper which are not very relevant to our manuscript.
The following are the 8 incorporated new research paper:
West, J.S.; Holdgate, S.; Townsend, J.A.; Edwards, S.G.; Jennings, P.; Fitt, B.D.L. Impacts of changing climate and agronomic factors on fusarium ear blight of wheat in the UK. Fungal Ecology. 2012, 5, 53-61.
Wilson,W.; Dahl, B.; Nganje, W. Economic costs of Fusarium Head Blight, scab and deoxynivalenol. World Mycotoxin Journal. 2018, 11, 291-302.
Khan, M.K.; Pandey, A.; Athar, T.; Choudhary, S.; Deval, R.; Gezgin, S.; Hamurcu, M.; Topal, A.; Atmaca, E.; Santos, P.A. Fusarium head blight in wheat: contemporary status and molecular approaches. 3 Biotech. 2020, 10, 172.
Kryszczuk, I.P.; Solarska, E.; Wiater, M.K. Communication Reduction of the Fusarium Mycotoxins: Deoxynivalenol, Nivalenol and Zeara-lenone by Selected Non-Conventional Yeast Strains in Wheat Grains and Bread. Molecules. 2022, 27, 1578.
Liao, Y.X.; Peng, Z.; Chen, L.K.; Nüsslerc, A.K.; Liu, L.G.; Yang, W. Deoxynivalenol, gut microbiota and immunotoxicity: A potential ap-proach? Food Chem Toxicol. 2018, 112, 342-354.
Zheng, X.F.; Zhang, X.L.; Zhao, L.N.; Apaliya, M.T.; Yang, Q.Y.; Sun, W.; Zhang, X.Y.; Zhang, H.Y. Screening of Deoxynivalenol Pro ducing Strains and Elucidation of Possible Toxigenic Molecular Mechanism. Toxins (Basel). 2017, 9, 184.
Shostak, K.; Bonner, C.; Sproule, A.; Thapa, I.; Shields, S.W.J.; Blackwell, B.; Vierula1, J.; Overy, D.; Subramaniam, R. Activation of bio-synthetic gene clusters by the global transcriptional regulator TRI6 in Fusarium graminearum. Mol Microbiol. 2020, 114, 664-680.
Chen, Y.X.; Chen, Y.S.; Shi, C.M.; Huang, Z.B.; Zhang, Y.; Li, S.K.; Li, Y.; Ye, J.; Yu, C.; Li, Z. SOAPnuke: a MapReduce accelera-tion-supported software for integrated quality control and preprocessing of high-throughput sequencing data. Gigascience. 2018, 7, 1-6.
- It is recommended that the authors add a description of the novelty of the article in the introduction and abstract sections.
We thank the reviewer for this suggestion. We have discussed the novelty of the article in the introduction and abstract sections.
The experimental notes in this article are overly interesting, but the novelty of the article is not adequately described and the manuscript writing suffers from grammatical and formatting errors.
We have discussed the novelty of the article in the introduction and abstract sections. We have fixed the grammatical and formatting errors.

Round 2
Reviewer 2 Report
-